# Digital Help for Substance Users (SU): A Systematic Review

**DOI:** 10.3390/ijerph191811309

**Published:** 2022-09-08

**Authors:** Natale Salvatore Bonfiglio, Maria Lidia Mascia, Stefania Cataudella, Maria Pietronilla Penna

**Affiliations:** Department of Pedagogy, Psychology, Philosophy, Faculty of Humanistic Studies, University of Cagliari, 09123 Cagliari, Italy

**Keywords:** substance use, digital treatments, alcohol, marijuana/cannabis, amphetamines, cocaine, tobacco, opioids, heroin, benzodiazepines

## Abstract

The estimated number of Substance Users (SU) globally has currently reached a very high number and is still increasing. This aspect necessitates appropriate interventions for prevention and specific treatments. The literature shows that digital treatments can be useful in the context of health services and substance abuse. This systematic review focuses mainly on research on the effectiveness of digital treatments for SU. Data sources included studies found on PsycINFO, PubMed, SCOPUS, and WebOfScience (WOS) database searches. The following keywords were used: *TITLE (digital OR computer OR software OR tablet OR app OR videogame OR seriousgame OR virtualreality) AND ABSTRACT((mental AND health) AND (addiction OR dependence OR substance OR drug))*. We focused on peer-reviewed articles published from 2010 through 2021 using PRISMA guidelines. A total of 18 studies met the inclusion criteria (i.e., type of intervention, efficacy in terms of misuse of substances and scored outcomes from questionnaire or toxicology tests, study methodology). The studies included investigations of specific digital treatments for SU of various kinds of drugs. The interventions were administered using personal computers, smartphones, or, in a few cases, tablets. Most of the interventions focused on the cognitive behavior therapy (CBT) model and/or on the use strategies, tips, or feedback. A minority provided information or training programs. The current review shows that digital treatments and interventions are effective in reducing the frequency of use, augmenting abstinence, or reducing the gravity of dependence for most of the studies at post-treatment. However, due to the heterogeneity of the variables (i.e., substance type, digital tool used, and treatment administered), there was a reduced generalizability of the results. This review highlights the need to continue the research in this field, and above all, to create effective digital protocols.

## 1. Introduction

Digital treatments can be useful in the context of health services. Our attention is directed to digital treatments for mental health [1] and, especially, to dependence disorders [2,3,4,5]. There is increasing evidence of the expansion of digital tools and systems for assessment, intervention, support, and prevention in the field of mental health thanks to, in part, the evidence provided by many studies and scientific research on the effectiveness of these treatments in helping people with mental health disorders [6]. The convergence of digital technologies with the fields of mental health and health care, which we can refer to as digital health or *mHealth*, has allowed for the increasing development and diffusion of these technological tools, resulting in continuous innovation. This field is increasingly active and growing, and treatments range from screening and monitoring to diagnosis and therapies (often administered alongside traditional therapies) [7,8,9,10,11,12,13,14]. It makes use of ‘wearable’ devices—such as virtual reality helmets, patches, digital patches, and tattoos—that have innumerable screening and monitoring functions and make use of increasingly innovative technologies using artificial intelligence algorithms and nanotechnology. By contrast, ‘nonwearable’ technologies are based on the use of smartphone applications, computer software and programs, internet platforms with psychoeducational or therapeutic purposes, video games or training, avatars, or typical chat tools (such as chatbots) implementing artificial intelligence technologies [15]. The relationship between health and digitalization was strongly highlighted by the COVID-19 pandemic [16,17]. These unprecedented times led to a big ‘boom’ in the use of technology in the health and mental health fields, as many people experienced the difficulties of accessing services in the traditional way [18]. These technologies can help fill possible gaps due to difficult access to care, especially for younger people, who are more prone to seeking technology-based healthcare services. Digital interventions also have other advantages, such as availability at any day of the week and at any hour (especially if they do not require an operator, as in the case of chatbot interventions) and can be ‘on-demand’. In addition, many interventions have lower costs than traditional treatments. Finally, they can help people who do not access services in the traditional way due to shame and social stigma, thus increasing the number of people who can use them [19]. Illness has costs for the individual and society; therefore, it is necessary to find as many solutions as possible to promote health and increase access to health services (including digital healthcare solutions). Among health preservation services, special attention should be paid to addiction or substance abuse [4,20]. Although many studies and reviews [12,13] have focused on SU, more research is needed to prove the effectiveness of specific digital treatments.

This is the reason why in this systematic review we focus on the digital based treatments for SU [21,22]. SU is referred to habitual substance users and can also include substance use disorders. A body of literature [5,23,24,25,26] shows the benefit of using digital mode to reach people who have limited access to treatments or presents other limitations in accessing standard protocols. However, the field of digital interventions for treating the use of substances is vast and varied, and likewise, the interventions for such disorders, and the digital tools currently available. 

A principal point to consider in SU is that a dependent person lacks sufficient control over the use of a drug, thus acting with behavioral dyscontrol and in a discontinuum. A principal characteristic of dependence syndromes is the uncontrollable desire to use substances such as drugs, alcohol, or tobacco. The literature also underlines that in dependence syndromes, returning to substance use after a period of abstinence causes a more rapid reappearance of the features of the syndrome that occur in nondependent subjects [27].

In addition, SU sometimes are refractory to any type of treatment and have very high drop-out rates. Whereas in-person treatments have been validated and have provided effective protocols, digital treatments do not have such a long history, and very often, their protocols is not tied to the interface. If there are existing effective protocols, there needs to be more connection between those who make the content and those who create the technological instruments. Teamwork is needed to integrate the strengths of in-person interventions with those of digital interventions. There is also a lack of individualized treatments at the technology level. Hence, more focus is needed on this aspect.

This systematic review was made in order to better understand (i) the effectiveness of digital treatments for SU, (ii) results indicated in each study, and (iii) which kind of digital treatment is more effective in relation to the population of the study and to the specific substance used. Furthermore, this review was designed to provide a framework that would aid researchers in evaluating new literature, give new directions for future research, and help to create useful and shareable treatments with adequate digital support. 

## 2. Methods

### 2.1. Overview

This systematic review was conducted in accordance with the Preferred Reporting Items for Systematic Review and Meta-Analyses (PRISMA) guidelines [28,29].

PsycINFO, PubMed, SCOPUS, and Web of Science (WOS) databases were systematically searched, using the following keywords: TITLE (digital OR computer OR software OR tablet OR app OR videogame OR seriousgame OR virtualreality) AND ABSTRACT ((mental AND health) AND (addiction OR dependence OR substance OR drug)). The search strategy has been included as Appendix A. We focused on peer-reviewed articles published from 2010 through 2021. Results were limited to English, Italian, and Spanish language peer-reviewed journal publications. Primary searches were completed in December 2021.

### 2.2. Inclusion and Exclusion Criteria

To be included in the review, studies had to report treatments using digital technology (i.e., using a computer, APP, tablet, or smartphone) for SU. SU participants should have been selected either through formal diagnosis or self-identification of current or past problematic substance use. Outcomes have to be measured through validated and/or standardized questionnaires (e.g., Alcohol Use Disorders Identification Test or AUDIT) or through the Visual Analogue Scale, or questions asked by the authors of participants and directly related to SU. Studies were also included if they refer to participants with several psychiatric disorders, including SU.

Book chapters, meta-analyses, reviews, comments, letters, and theoretical papers were excluded. Studies were also excluded if they: (i) did not report a digital technology treatment for SU; (ii) considered problematic internet, app, or computer use; (iii) reported use of the tools (e.g., computer or tablet) to administer surveys; (iv) used Virtual Reality to reproduce the effects of drugs (e.g., hallucinations); (v) reported a standardization or validation of a treatment using an app or a serious game. Moreover, ad hoc, or non-validated instruments or indirect outcomes (e.g., depression or anxiety) were not considered as outcomes.

### 2.3. Study Selection and Extraction Steps

All identified citations were imported into the bibliographic manager software Zotero 5.0 (Corporation for Digital Scholarship, Vienna, VA, USA). Duplicates were identified and removed, after which abstracts, and titles were screened by two independent reviewers (MLM and NSB) for eligibility. Discordant eligibility determinations were resolved by consensus.

The full texts of the eligible records were then obtained and screened for eligibility according to the exclusion criteria. Any doubts or conflicts were resolved by discussion between the three reviewers (M.L.M., N.S.B., and M.P.P.), to reach a consensus.

### 2.4. Data Extraction

Two independent reviewers (M.L.M. and N.S.B.) created a data extraction standardized form in Microsoft Word (Microsoft Corporation, Redmond, WA, USA). The first author, year of publication, number of participants, diagnosis of participants, age, study setting, comparator, outcomes, type of treatments, and devices used were extracted from each included study. Any discrepancies in the extracted data were resolved by a third reviewer (M.P.P.).

### 2.5. Synthesis

The study quality and characteristics of interest were tabulated and narratively described. Two independent reviewers (M.L.M. and N.S.B.) assessed the quality of retrieved studies [30]. A standardized quality tool was used based on the following quality criteria (see Appendix A): a minimum of 50 subjects per sample, validated measures, follow-up, pre-post training design, randomized subject selection and/or condition, presence of control condition, presence of placebo condition, and evidence of utility.

Each criterion was recorded as 1 (present) or 0 (absent). Score ranged from 0 to 8. The higher the score the higher the quality of the study. 

## 3. Results

### 3.1. Study Selections and Extractions

A total of 530 abstracts were located, of which 325 were removed, being duplicates. A total of 205 studies were screened against titles and abstracts. Subsequently, 139 studies were excluded, and among the 66 studies assessed for full-text eligibility, 48 studies were excluded. A total of 18 studies finally met our inclusion criteria and were included (Figure 1). In the final selection, only English studies were included.

### 3.2. Study Characteristics

We summarized the key results of the study characteristics in Table 1.

A sample size of a total of 25,475 subjects were involved in all the selected studies, of which 7161 were male, 9216 females, and 17 were classified as other gender categories. Two studies did not report gender numerosity. The average age for the participants was 40.9. For most of the studies, the principal substance selected for treatment was alcohol (*n =* 12), followed by marijuana/cannabis (*n =* 8), stimulants (e.g., amphetamines or cocaine) (*n =* 7), tobacco (*n =* 5), opioids (*n =* 4), heroin (*n =* 4), other drugs (*n =* 3), and benzodiazepines (*n =* 2). One study did not specify the type of drugs selected.

The majority of the interventions were administered through a personal computer (PC)—i.e., with an on-line platform (*n =* 14). The use of smartphones (i.e., app for smartphones) was less common (*n =* 5), as well as tablets (*n =* 2).

Moreover, the selected studies reported a heterogeneity of the type of approaches and models used for digital treatment. The majority used Cognitive Behavioral Therapy (CBT) (*n =* 7), followed by Brief Alcohol Intervention (BAI) (*n =* 2), Informative intervention (*n =* 2), Attentional Bias (*n =* 2), Interactive Voice Response (IVR) (*n =* 1), Digital Recovery Support Services (D-RSS) (*n =* 1), e-Learning (*n =* 1), Computer-Guided Therapy (CBI) (*n =* 1), Self-Determination Therapy (SDT) (*n =* 1), Health Action Process Approach model (*n =* 1), Effectiveness of Computer-tailored Smoking Cessation Advice in Primary Care (ESCAPE) (*n =* 1), and the Information-Motivation-Behavior model (*n =* 1).

Substance use outcomes were measured with validated measures (*n =* 13), or as a means of frequencies of substances use (e.g., days of use, quantity of use) (*n =* 11), abstinence (e.g., days of abstinence) (*n =* 3), recovery measures (e.g., functioning in domains that are implicated in SUDs) (*n =* 5), or with biological tests (e.g., urine test) (*n =* 3).

As regards methodology and study design, most of the selected studies used follow-up assessment (*n =* 16) or a pre-post evaluation after intervention (*n =* 2). Fewer studies have divided samples in subgroups (e.g., separate conditions) (*n =* 10). Of these, nine studies have randomized conditions and seven studies have added control groups/conditions. No studies used a placebo condition.

As regards the results, 17 out of 18 studies indicated obtaining positive results at least for one of the outcomes, as a means of reduced abstinence, days of use, or reduced gravity of addiction. However, only four out of nine studies reported differences of utility between groups or conditions, and three studies did not compare groups or conditions (see Appendix A).

## 4. Discussion

This paper aimed to detect effectiveness of existing digital tools in reducing substance use. We used restricted criteria to identify relevant studies and report (i) the type of intervention, (ii) its efficacy in terms of substance misuse and scored outcomes from questionnaire or toxicology tests, and (iii) the study methodology.

Despite the optimism surrounding the use of digital interventions to reduce the misuse of substances such as alcohol and tobacco, the evidences presented on the effects of such interventions are weak. Digitally delivered interventions for substance misuse are likely to require robust evidence of their effectiveness if they are to be widely adopted and compared to real-world interventions and settings.

### 4.1. Outcomes and Measurements

Overall, 17 out of 18 studies reported at least one positive outcome for the reduction of substance misuse in the evaluated population. Although the study quality and data analysis were generally weak, the results suggested that digital interventions may produce some reduction in substance misuse. However, not all of the selected studies measured substance misuse or the gravity of dependence as primary outcomes.

We found that digital treatments and interventions are effective in reducing substance misuse—more precisely, decreasing the frequency of use, augmenting abstinence, or reducing the gravity of dependence in most of the studies at post-treatment.

In addition, most studies have demonstrated treatment effectiveness as a function of frequency measures, such as abstinence (*n =* 2) or the reduction in the amount of substance use (*n =* 9), rather than standardized (*n =* 3) or objective (*n =* 1) measures. It would be desirable, in our opinion, to prove the effectiveness of treatment using primarily standardized instruments that measure the severity of addiction and use objective instruments (e.g., biomarkers). In fact, some of the studies selected in this review reported, for example, a reduction in the frequency of use and in scoring scores, albeit not always statistically significant.

Moreover, most of the studies also measured treatment effectiveness for only one substance. A treatment effect on the main substance (reported by the subject) does not imply abstinence from other substances. In fact, the literature shows that people are more likely to use more than one substance (poly-dependence) rather than one (mono-dependence), resulting in increased severity of dependence and difficulty in treatment. Therefore, to obviate possible bias in the results, one should measure the subject’s level of addiction rather than the use of a specific substance.

Finally, only two studies used cognitive bias as a measure of dependence, and only one found a reduction in bias at follow-up. Cognitive bias (e.g., the Stroop effect) is a very reliable proxy measure because it is an objective measure of a subject’s level of addiction. The use of a proxy measure, such as a cognitive bias (widely used in the literature), could increase the reliability of the effectiveness of a digital treatment. The variation in outcome measures and the main use of frequency and abstinence as outcomes—instead of more objective measures—reflects the variability of the aims of the selected studies (e.g., evaluating the cost effectiveness of an intervention), in which substance misuse was considered merely a secondary outcome. It should be noted that this could also reflect the lack of a gold standard for measurements [49], as mentioned by some of the authors in the limitations section of their study [44], despite the existing literature on some of the most studied substances [50,51,52]. A few studies have used an alternative and more objective measure to cover this deficiency: toxicology screening tests [32,42,45]. Nevertheless, toxicology screenings have their own disadvantages, such as the inability to detect mild use and the lack of privacy—users might prefer to avoid face-to-face contact with professionals [2,3].

A minority of the selected studies computed the effect size [38,39,40] and found low or medium effect sizes. If we compare the effect sizes resulting from digital interventions, as reported in this review, with those of real-world interventions, we observe that real-world treatment interventions produce not only small or medium but also large effect sizes [53,54]. Therefore, we suggest that future studies investigate novel approaches to increasing the effect sizes of digital interventions.

### 4.2. Models and Approaches

Most of the interventions focused on the CBT model and/or on the use of strategies (i.e., coping strategies), tips, or feedbacks. A minority provided information or training programs (i.e., attentional bias). However, the variability of models and approaches on which the interventions in the selected studies were based and the presence of very few studies reporting multi-interventions-combinations of digital intervention [31,34,36] and comparisons with the control condition (see Appendix A) reduce the generalizability of the determined utility and effectiveness of the strategies used.

Regarding the effectiveness of treatments, those based on continuous programs to be carried out in different steps, tailored with exercises and training, seem to be more effective than short or spot interventions. For example, some teach how to use coping strategies and require the subjects to follow a program over a certain period, some engage subjects in psychoeducation or behavioral change programs (such as CBT) structured over a long period, while others are tailored to the patients’ needs and provide counseling forums and materials. In our opinion, greater effectiveness might be achieved by (a) medium- and long-term treatments requiring continuity from the subject or (b) by having a solid theoretical basis that has been proven to be effective in treating addiction. From this point of view, it might be the treatment itself that is already effective, regardless of whether digital technology is used or not.

### 4.3. Methodology and Intervention Design

Only three studies were able to blind personnel [31,36,52] and no studies reported blinding participants to ensure that they were unaware of the other conditions and that the outcome measures were unaffected by possible knowledge of the received intervention. This leads to a high risk of performance and detection bias that, however, depends on the nature of the digitally delivered behavioral interventions in most of the included studies.

Follow-up assessments indicated that the post-treatment effects were sustained for up to three months from the interventions. This indicates that digital intervention is a suitable approach to achieve a lasting small reduction in substance use. However, many studies evaluate interventions within 12 months and only one study evaluates intervention over 12 months [33]. Future research should aim to evaluate the effectiveness of digital intervention programs beyond three years to better understand how program effects can be sustained. Moreover, only a few studies compared and reported differences between the groups, despite having divided their samples into subgroups. This could be a methodological weakness leading to nongeneralized results.

However, the moderate successes of digital interventions support the notion that interventions for substance misuse may have an impact of utility. Although not the focus of this study, it has been widely acknowledged that digital interventions are more likely to be successful in populations that have played an active role in their design [55,56]. Therefore, future interventions should employ a centered design and intervention (UCDI) approach throughout their design, development, and evaluation to elicit their potential in substance misuse reduction. To pursue this aim, it is essential that the core user is carefully defined in the design of these digital interventions to increase the likelihood of addressing user needs and expectations. This may be achieved through techniques such as persona building, storytelling, and role playing [57]. User engagement and the acceptability of an intervention are crucial to its success, indicating that UCDI processes may be a way to increase the effectiveness of digital interventions and should be considered integral to the evaluation process. The use of a UCDI is also important if we assume that most people using drugs (e.g., club drugs) do not want to engage with traditional treatments or support services for fear of stigma and due to concerns about relevance [58,59]

## 5. Conclusions

The effectiveness of existing digital treatments for reducing substance misuse among SU in a wider population was reviewed. Unfortunately, the overall quality of the 18 included studies was weak; thus, definitive conclusions regarding the effectiveness of these interventions could not be drawn.

There is a clear lack of studies with long-term follow-ups (more than 12 months), control conditions, randomized samples, and blinded conditions. Moreover, there is a lack of evidence for other populations, such as employees and ethnic minority groups. Future primary studies and reviews should include these aspects. In addition, most of the useful results were obtained with self-assessment outcomes and measures, such as frequency of use and days of abstinence. Useful results on validated addiction assessment instruments or through objective tools such as biomarkers and cognitive bias would certainly have given greater reliability to the effectiveness of the treatments used.

A good portion of the studies selected for this review still provided useful results. However, it is possible that these results depended mainly on the use of theoretical models and techniques whose effectiveness are already known from literature. The lack of placebo conditions and comparisons with control conditions represents another weak point that does not help to determine how effective a digital intervention is.

In order to increasingly improve digital health interventions, it is desirable that authors and creators of digital health tools increasingly involve users in the construction of the intervention itself and in the design and guidelines of APPs and software. For example, training and serious games (i.e., serious video games) seem to play a particularly significant role in the field of *mHealth*. Especially serious games are becoming increasingly relevant. For years, the video game industry has been developing products that not only entertain but are also educational, pedagogical, and therapeutic, focusing on learning, memory, and rehabilitation, and so on, enabling them to reach an increasingly broad audience. In the field of rehabilitation, for example, they make it possible to activate a dynamic process of adaptive and programmed change in the user, in response to an unplanned change due to a problem, disorder, or trauma that the individual presents. The possibilities offered by treatments that use technologies such as serious games and training administered through tablets or PCs are their functionality, their use even at a distance, and above all the possibility of activating a process of short- and long-term changes, thanks to the continuous and constant use of the serious game. The effectiveness of this type of intervention is, thus, in line with the transformations taking place in the field of care services. Certainly, what is crucial is to continue to pursue research in this area that can ensure comprehensive services that are accessible to all, can be functional, effective, reach the greatest number of users, and can be shareable.

In this sense, it is necessary to design studies that involve users, health professionals, designers, and developers to have truly applicable programs for reducing substance abuse and addiction.

However, the results showed in this review suggest that the success of digital interventions may not only be confined to the misuse of illicit substances, but also to other aspects, such as the changing of strategies, self-regulation, and behavioral mechanisms delivered by the interventions [60,61]. Therefore, while digital interventions are a promising development area, it is important that interventions undergo robust creation and evaluation processes, and that effective implementation strategies be used that are best suited to their context [62].

## 6. Limitations

This study has several strengths and limitations which should be noted. Even though a rigorous search criterion was used, few databases were considered for searching. Moreover, we did not include a gray literature search and reference list screening; thus, relevant studies were potentially missed. The main use of only two reviewers throughout the screening and quality appraisal processes could have led to the risk of bias. Finally, we did not undertake a rigorous quality assessment of the reviews.

## Figures and Tables

**Figure 1 ijerph-19-11309-f001:**
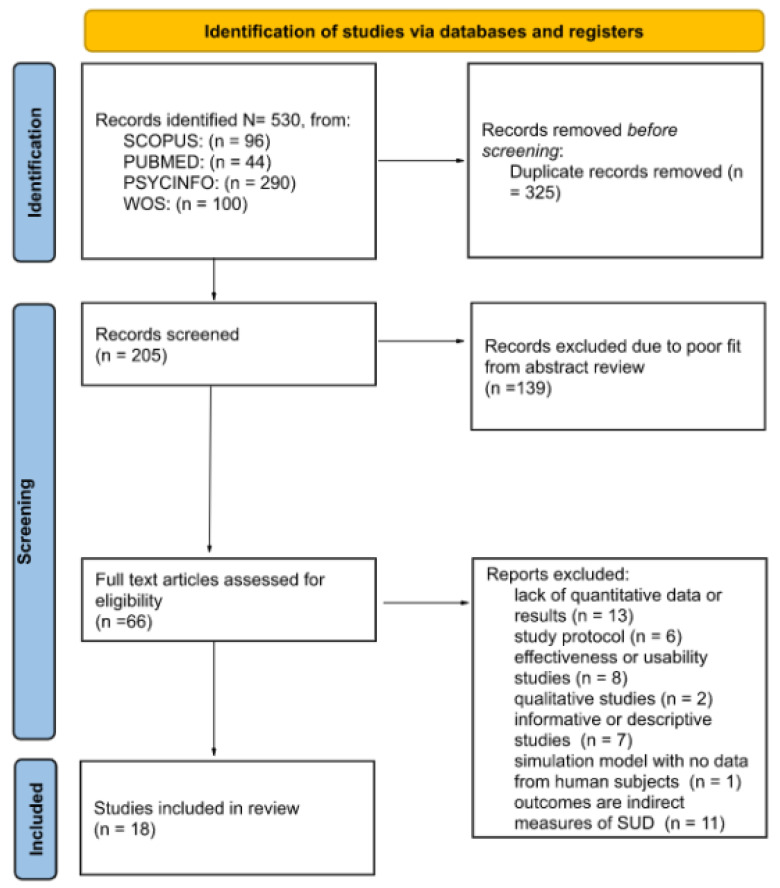
Systematic literature search, screening, and relevance assessment conducted according to PRISMA guidelines.

**Table 1 ijerph-19-11309-t001:** Characteristics of publications of treatment(s) with digital technology included in review (N = 18).

Study	Participants (n), Mean Age (SD), Characteristics, Type of Dependence/Abuse	Tool/s	Outcome(s) and Measure(s)	Treatment(s) with Digital Technology	Results
Chander et al., 2021 [31]	*n =* 439, women; median age = 31 years; women were randomized as follows: 146 were assigned to the CBI + IVR/Text group, 145 to the CBI only group, and 148 to the control group; type of dependence: alcohol.	PC	Self-reported alcohol consumption Alcohol biomarker phosphatidylethanol (PEth) test. Heavy drinking days, drinking days, drinks per drinking day, and drinks per week. The Alcohol Use Disorders Identification Test (AUDIT)—only at baseline. Daily use of illicit substances using the 30-day timeline follow-back.	*Computer-delivered brief alcohol intervention (CBI).* This interactive 20-min intervention was delivered in a motivational interviewing style by a three-dimensional avatar, using the Motivational Enhancement System (MES) Platform. CBI was delivered using interactive voice response technology (IVR) and text messages.	*Assessments at baseline and 3, 6, and 12 months.* Participants in all three study conditions significantly reduced their heavy drinking days, drinking days, drinks per drinking day, and drinks per week over the follow-up period (*p* ≤ 0.001), with no statistically significant differences between study conditions. CBI with or without IVR+text messages did not results in greater reduction in alcohol use compared to the control group.
Cucciare et al., 2021 [32]	*n =* 138, veterans; mean age = 63.18 years; men were randomized as follows: 71 in standard intervention and 67 in CBI intervention; type of dependence: alcohol.	PC	Number of drinking days and number of days engaging in unhealthy drinking in the past 30 days. The number of standard drinks consumed per drinking day as a secondary outcome.	*Computer-delivered brief alcohol intervention (CBI), as described previously.*	*Subjects were tested at baseline and 3- and 6-month follow-up.* Participants in the CBI condition reported significantly fewer drinking days and unhealthy drinking days than participants enrolled into the standard care condition (*p* ≤ 0.05). Participants in the CBI condition reported significantly fewer unhealthy drinking days at 3-month follow-up compared to participants in the standard care condition (*p* ≤ 0.05).
Curtis et al., 2019 [33]	*n =* 729 women; mean age = 46.83 years; type of dependence: alcohol, opioids, heroin, benzodiazepines, cocaine, amphetamine, marijuana.	Digital recovery support service accessing on-line	Primary recovery pathway from a list of mutually exclusive options (e.g., abstinence-based 12-step, abstinence-based non-12-step, and medication).	*SHE RECOVERS (SR) as a Digital Recovery Support Services (DRSS).* It is a digital community that includes a public Facebook page, two private Facebook groups, digital training events, digital recovery coaching, a website, and an email listserv.	*Subjects were tested at 1, 1 to 5, and 5+ years.* Participants of SR community and other DRSSs with less than 1 and 1 to 5 years in recovery reported pathways of abstinence-based 12-step mutual aid at higher rates (*p* ≤ 0.001).
Danaher et al., 2019 [34]	*n =* 1271 participants; mean age = 44.9 years; participants were randomized as follows: 633 on MobileQuit and 638 on QuitOnline; type of dependence: nicotine.	Mobile app (for smartphone) and nonmobile PC	Number of smoked cigarettes. Tobacco history (years of use, number of quit attempts, and amount of use). Fagerstrom Test for Nicotine Dependence. Alcohol use assessed at baseline using a single item. Seven-day point prevalence use of cannabis.	*The MobileQuit intervention optimized for smartphones.* *QuitOnline intervention designed* *primarily for use on mobile PCs.* These two interventions present very similar best practice smoking cessation content based on cognitive behavior therapy (CBT) features.	*Participants were screened at baseline and 3 and 6 months.* At 3 (*p* ≤ 0.001) and 6 (*p* = 0.02) months, participants in the MobileQuit condition displayed greater smoking abstinence than those in QuitOnline and used repeated point prevalence at 3 and 6 months (*p* ≤ 0.001). MobileQuit participants displayed greater smoking abstinence at 3 months (*p* ≤ 0.001) and at both 3 and 6 months (*p* ≤ 0.001), but not at 6 months.
de Ruijter et al., 2019 [35]	*n =* 269 practice nurses; mean age = 47.3 years; participants were randomized, 147 in the intervention group and 122 in the control group; type of dependence: nicotine.	PC	Fagerström test for nicotine dependence (FTND). Smoking abstinence.	*Computer-tailored e-learning program.* It consisted of five e-learning modules with tailored advice, a forum, and smoking cessation counseling materials; three general modules containing project information, frequently asked questions about the trial, and a counseling checklist to self-report application of guideline steps.	*Tests were administered at baseline and at 6 and at 12 months.* A significant difference was found at 6 months on the FTND (*p* = 0.01), reporting a lower score as a means of reducing dependence for the intervention group, compared to the control group.
Drislane et al., 2020 [36]	*n =* 780 patients aged 18 to 60; participants were randomized as follows: 266 in the Therapist-Delivered Brief Intervention (TBI), 257 in the Computer-Guided Brief Intervention (CBI), and 257 in the enhanced usual care (EUC) as control group; type of dependence: alcohol and cannabis.	PC	Alcohol Use Disorders Identification Test (AUDIT). Cannabis use frequency as measured by the National Survey on Drug Use and Health (NSDUH).	*Therapist-delivered brief-intervention (TBI).* *Computer-guided brief-intervention (CBI).* Intervention with TBI and CBIs involved touchscreen-delivered and audio-assisted content. The TBI was administered by a Master’s-level therapist, whereas the CBI was self-administered using a virtual health counselor.	*Assessment was administered at baseline and 3, 6, and 12 months.* There was a significant reduction in cannabis use over time in the TBI group (*p* ≤ 0.05), but not in the EUC group. Only participants aged 18 to 25 years who received TBI showed significant reductions in cannabis use. Moreover, the reductions in alcohol use after TBI were found among men (*p* ≤ 0.01), but not among women. Although CBI reduced cannabis use days when examined as a sole outcome, it did not result in significant reductions in severity of alcohol use and cannabis use relative to EUC.
Elison-Davies et al., 2020 [37]	*n =* 5792 individuals; mean age = 40.54 years; a total of 1489 (26%) participants provided post-treatment data; type of dependence: nicotine, alcohol, opioids, heroin, benzodiazepines, cocaine, amphetamine, cannabis, novel psychoactive substances, prescribed medications.	on-line PC	Questions regarding drug/alcohol consumption and drug/alcohol consumption goals Severity of Dependence Scale (SDS). Recovery Progression Measure (RPM), measuring functioning in six biopsychosocial domains implicated in drug misuse and recovery.	*Breaking Free Online (BFO).* It is a digital intervention for individuals with substance misuse, containing 12 main behavioral change techniques that can be delivered with practitioner support as “computer-assisted therapy” or as self-help. The BFO program uses baseline RPM data to populate a visual depiction of a six-domain biopsychosocial model, the “Lifestyle Balance Model” (LBM), which forms the theoretical underpinnings of BFO and is based on the five-factor model used in cognitive behavioral therapy (CBT).	Effect sizes estimation revealed a medium effect size for changes in self-reported weekly alcohol consumption (*r* = 0.55), and small effect sizes for changes in self-reported drug consumption (*r* = 0.47), and severity of drug (*r* = 0.29) and alcohol dependence (*r* = 0.28). Significant reductions in SDS score and in overall RPM were also found (*p* ≤ 0.001).
Elison-Davies et al., 2021a [38]	*n =* 2571 individuals; mean age = 38.42 years; a total of 1107 (43%) completed a post-treatment assessment; type of dependence: heroin.	on-line PC	Severity of Dependence Scale (SDS). Recovery Progression Measure (RPM) measures functioning in six biopsychosocial domains that are implicated in substance use disorders SUDs.	*Breaking Free Online (BFO)*, as described previously.	*Participants were provided with access to the computer assisted treatment program for 12 months, and engaged with it as self-directed treatment.* A medium effect size was found for reductions in weekly opioid use (*r* = 0.71), and small effect sizes for reductions in severity of opioid dependence (*r* = 0.42) from baseline to post-treatment. Improvements were also found in all RPM six domains (*p* ≤ 0.001).
Elison-Davies et al., 2021b [39]	*n =* 1830 participants; mean age = 33.80 years; a total of 460 subjects (25%) completed both at baseline and at follow-up assessment; type of dependence: opioids.	on-line PC	Severity of Dependence Scale (SDS). Recovery Progression Measure, (RPM) measures functioning in six biopsychosocial domains that are implicated in SUDs.	*Breaking Free Online (BFO)*, as described previously.	*Participants were provided with access to the computer assisted treatment program for 12 months, and engaged with it as self-directed treatment.* Differences with small effect sizes were found among baseline and follow-up measures of cannabis use and RPM (*r* = 0.30 to 0.48; *p* ≤ 0.001).
Elison-Davies et al., 2017 [40]	*n =* 2311 individuals; mean age = 42.2 years; type of substances: heroin, cocaine, alcohol, prescribed and substitute medications, cannabis, amphetamines, novel psychoactive substances, tobacco, and club drugs.	on-line PC	Severity of Dependence Scale (SDS). Recovery Progression Measure (RPM) measures functioning in six biopsychosocial domains that are implicated in SUDs.	*Breaking Free Online (BFO)*, as described previously.	*Participants were provided with access to the computer assisted treatment program for 12 months and engaged with it as self-directed treatment.* The psychometric assessment was repeated at a mean of 8.2 weeks from baseline. Medium effect sizes were identified for reductions in alcohol and drug dependence between baseline and follow-up (*r* = 0.51). Smaller effect sizes were identified for changes in scores for RPM between baseline and follow-up (*r* = 0.19 to 0.39). Changes in severity of alcohol dependence was associated with completion of some LBM strategies, specifically “lifestyle” (*p* ≤ 0.012) and “negative thoughts” (*p* ≤ 0.009). Changes in scores for drug dependence were not associated with the number of times participants completed strategies in the six LBM modules (*p* ≤ 0.051). Changes in total RPM were associated with the number of times participants completed LBM module strategies, specifically on the “negative thoughts” module (*p* ≤ 0.001).
Han et al., 2018 [41]	*n =* 75 participants; mean age = 41.6 years; subjects were randomized as follows: 50 in the experimental group and 25 in the control group; type of substances: heroin, amphetamine-type substances (ATS).	smartphone app	Daily situations (e.g., drug use, craving, and coping) were collected by the daily survey, which was conducted every day at a scheduled time through an Ecological Momentary Assessment (EMA). Life Experience Timeline Assessment (LET) questionnaire that assesses 20 events (e.g., substance use, emotion, coping, and craving) over the past week. Urine test that identified heroin, ATS, marijuana, cocaine, and ketamine use.	*mHealth app, developed specifically to help individuals with SUDs achieve and maintain recovery.* The mHealth app is based on cognitive behavior therapy (CBT), which emphasizes triggers and coping strategies for relapse prevention, and self-determination theory (SDT), which motivates people to change and act for themselves.	*Drug use results were provided at week 1 (W1), 2 (W2), 3 (W3), and 4 (W4).* The number of subjects of the experimental group using/not using substances for each week provided by urine test, LET, and EMA were as follows. *Urine* use: W1 = 24; W2 = 21; W3 = 15; W4 = 11 not use: W1 = 19; W2 = 22; W3 = 27; W4 = 31 *LET* use: W1 = 15; W2 = 12; W3 = 10; W4 = 7 not use: W1 = 33; W2 = 36; W3 = 38; W4 = 41 *EMA* use: W1 = 12; W2 = 10; W3 = 6; W4 = 5 not use: W1 = 28; W2 = 25; W3 = 26; W4 = 25
Kay-Lambkin et al., 2014 [42]	*n =* 35 clients; mean age = 42.11 years; subjects were divided as follows: 12 exposed to SHADE and 23 not exposed; type of substances: alcohol and cannabis.	PC	Opiate Treatment Index (OTI) to assess the quantity and frequency of use for 11 different drugs.	*Self-Help for Alcohol and Other Drug Use and Depression (SHADE).* It incorporates cognitive behavioral therapy (CBT) strategies to encourage reductions in depression and AOD (Alcohol and other drugs) use.	*Client assessment was collected at baseline and at 15-week follow-up.* For alcohol use between baseline and 15-week follow-up assessment. Participants who did not receive the SHADE modules reported a three-standard-drink per day reduction and three-standard-use of cannabis per day reduction between baseline and at 15-week follow-up assessment. Participants who were exposed to SHADE reported an eight-standard-drink per day reduction and nine-standard-use per day reduction in cannabis use over the same time period.
Leightley et al., 2018 [43]	*n =* 150 individuals who served in the UK military; age = 18 to 65 years; type of substances: alcohol.	Smartphone	Alcohol consumption. Alcohol use disorders via alcohol use disorders identification test (AUDIT)	Alongside the app (InDEx app). This app uses daily automated personalized text messages (SMS), corresponding to specific behavior change techniques, with content informed by the Health Action Process Approach (HAPA) for the intended purpose of promoting the use of the drinks’ diary, suggesting alternative behaviors, and providing feedback on goals setting.	*Participants completed tests and measures at registration, on days 8, 15, and 22.* Participants reduced the alcohol consumption for the following outcomes per week (W): *drinking days (*):* W1 (media*n =* 4); W2 (media*n =* 3); W3 (media*n =* 3); W4 (media*n =* 3) *drink free days:* W1 (media*n =* 3); W2 (media*n =* 3); W3 (media*n =* 3); W4 (media*n =* 3) *unit per drinking days:* W1 (media*n =* 5.6); W2 (media*n =* 6.5); W3 (media*n =* 4.5); W4 (media*n =* 4.7) *unit consumed:* W1 (media*n =* 22.9); W2 (media*n =* 20.4); W3 (media*n =* 18.1); W4 (media*n =* 15.9) *alcoholic drinks per drinking day:* W1 (media*n =* 2); W2 (media*n =* 3); W3 (media*n =* 2); W4 (media*n =* 2) *binge drinking per day per week:* W1 (media*n =* 2); W2 (media*n =* 2); W3 (media*n =* 1); W4 (media*n =* 2) A small change in AUDIT score was observed for participants who self-reported for Day 0 (registration) and Day 28 (final day) based on median score.
Wernette et al., 2018 [44]	*n =* 50 pregnant women at risk for substance use; mean age = 23.3 years; women were randomized as follows: 31 allocated to the intervention condition and 19 allocated to control; type of substances: alcohol and marijuana.	PC	Self-report of illicit drug use. Hair sample testing (Psychemedics, Inc.) at baseline and at follow-up assessment to corroborate self-report of illicit drugs use.	*Innovative computer-delivered intervention (the Health Checkup for Expectant Moms, HCEM) that targets women at risk for STI/HIV and alcohol/drug use during pregnancy.* HCEM is a tailored, motivationally focused STI/HIV and substance use risk reduction intervention, and provides training in several relevant skills, informed by the Information-Motivation-Behavior (IMB) model, which theorizes that information and motivation activate one’s behavioral skills, which in turn lead to risk reduction.	*Participants were tested at baseline and at 4-month follow-up.* Women in the HCEM condition, compared to controls, had a significantly larger reduction in the odds of any self-reported marijuana or alcohol use from baseline to follow-up (*p* ≤ 0.015). The odds of alcohol or marijuana use at baseline were 11.7 times higher at baseline, compared with follow-up in women assigned to HCEM (*p* ≤ 0.001). Of the valid 27 hair samples, 5 were positive for cocaine (all were in the intervention condition), 1 of whom was also positive for opiates, and an additional 3 were positive for marijuana (1 control and 2 intervention).
Wodarski et al., 2015 [45]	*n =* 5775 college students; type of substance: alcohol and other substances not well specified.	computer-based intervention (merely informative).	Frequencies of drinking	The intervention provides college students with basic knowledge concerning substance use and abuse, and increases students’ awareness of their own potential risks by giving immediate feedback and individualized recommendations.	Binge drinking has dropped to 27% on campus (48% to 35% reduction in number of student reporting drinking five or more drinks at a time), and frequent binge drinking has dropped to 44% (25% to 14% reduction in the number of students reporting drinking five or more drinks at a time three or more times in the past 2 weeks).
Wu et al., 2014 [46]	*n =* 6911 adult smokers; age = 18 to 65 years; the complete case analysis included 3309 participants, randomized as follows: 1795 in the control group and 1514 in the intervention group; type of substance: nicotine.	PC	Measure of abstinence	*Effectiveness of computer-tailored Smoking Cessation Advice in Primary Care (ESCAPE), lasting 6 months.*	*Participants were tested at baseline and at 6-month follow-up.* The clinical results showed that the intervention produced a modest increase in quit attempts during the 6-month follow-up compared with the control group (Odds Ratio = 1.13). There were no significant differences in 3-month prolonged abstinence between the treatment groups at the 6-month follow-up.
Zhang et al. 2019 [47]	*n =* 30 individuals; mean age = 43.76 years; type of substances: opioids, alcohol, cannabis and stimulants	app	Addiction Severity Index (ASI)-Lite (retained only the drug and alcohol use questions). Severity of Drug Dependence Scale (SDS). Craving with a Visual Analogue Scale (VAS). The presence of attentional biases was determined, based on the mean reaction times taken to respond to the position of the probes that replace drugs or neutral stimuli.	*A mobile-based attention bias modification intervention: Visual probe.*	*Tests and questionnaires were administered only at baseline. Attentional bias was administered on pre- and post-training.* On day 1 of the intervention, participants were required to complete both a baseline attention bias assessment task and an attention bias modification training task (intervention). On the subsequent days (days 2 to 7), they completed the attention bias modification training task. Fourteen participants had positive attentional biases at baseline. For these, there was a general decrease in the attention bias scores from baseline to the end of the planned intervention trials. The changes in the scores ranged from 12 to 409.5 milliseconds, comparing the final attention bias scores (upon the completion of the intervention) with the baseline scores (at the start of the intervention).
Zhu et al. 2018 [48]	*n =* 40 subjects; mean age = 33.88 years; subjects were randomized as follows: 20 assigned to a computerized cognitive addiction therapy (CCAT) and 20 to a control group; type of substance: methamphetamine.	iPad	Methamphetamine addiction Stroop task was applied to measure the methamphetamine-related attentional bias.	*Methamphetamine Attention Bias Modification*. Participants included in the CCAT group were also undergoing standard treatment; in addition, the participants received the CCAT training program that lasted for 4 weeks (20 sessions, five times a week, each session lasting approximately 60 min). After every CCAT session, a 5-min relaxation session was carried out by playing light music and watching pictures with relaxing effects.	*Attentional bias was administered on pre- and post-training.* There were no significant differences between the two groups in attention bias.

* = median IQR (interquartile range).

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
