# Peer review of "Digital Help for Substance Users (SU): A Systematic Review"

_ijerph, 2022, doi:10.3390/ijerph191811309_

Round 1

Reviewer 1 Report

In the first sentence in your article, you say that "There is evidence that digital treatments can be useful in the context of health service,
in mental health[1], and in dependence disorders [2-5]".
So could you explain the originality of your study.

It might be interesting to make a paragraph showing    
that digital treatments can be useful in the context of health and addiction.

The study is very limited, it may be interesting to extend the study to take into account other studies and articles.

It may be interesting that the authors better explain which types of digital applications can be useful for treatment (video games, programming, etc).   

The study may be interesting if the authors have done their own survey study with patients to have a reliable result.

Explain better your methodology, your choice for the articles to be studied and your criteria (page 3 and 4).

How did the health workers (health care team) receive your result? Did they agree or disagree? Give a summary of their opinions.

Do you have any medical specialists on your team?

Table 1 is very difficult to read and understand, it should be improved.

Improve the parts: discussion, conclusion, etc.

Improve the language of your article.

Author Response

Thank you for your letter Ref. Manuscript ID: IJERPH -1832629-R1, entitled “Digital help for substance users (SU): A systematic review”, and for giving us the opportunity to review and resubmit the paper.

We are very grateful to your and the reviewers’ comments and suggestions; we are deeply appreciative of your careful reading.

Detailed replies to your comments are enumerated below, with the list of modifications and integrations. We hope this revised version now satisfies the requirements for publication in your journal.

Then, we submit the revised version of paper; for clarity new portions, added or modified in response to the referees’ comments, are highlighted in the manuscript; furthermore, the tracked version of the manuscript is attached.

Thank you very much

The Authors

Reviewer 2 Report

The topic discussed in the review is an interesting one, and the review is generally well conducted. However, there are a number of areas the manuscript could be improved:

 1) Introduction: A number of statements in the introduction do not have references to support them. There are some minor typos in the introduction and some words that are the wrong tense. Additional proof reading could address this. The aims of the review at the introduction are vague, and it would help the reader significantly if they were more clearly stated. I would suggest stating that the primary aim of the review was to determine X. Secondary aims of the review are Y, Z etc

2)     Methods: While the process used to analyse the review data was robust, there is a large amount of detail missing from the methodology that would enable the review to be replicated. It would be particularly useful to include more detail about the search strategy used, either in-text or as a Table detailing one of the database search strategies. At minimum it would be useful to know the exact keywords used to search for different technologies and different substance use disorders, as these terms would influence the extent to which any literature was missed in the review. It would also be beneficial to add more detail about the data synthesis, particularly the process used to determine the study quality. 

3)     Results: It would improve this section if sub-headings were included to break up the text. I would suggest adding sub-headings such as Substance Use Disorders or Population; Methodology and Intervention design; and Study Outcomes. It would also strengthen the article if more detail was added about the outcomes of studies in the results, as it seems to have been an aim of the review to determine the impact of digital interventions in substance use. 

4)     Conclusions and Discussion: The text under the conclusions sub-heading would be more appropriate in the Discussion section. The text under ‘Future Directions’ is more like conclusion text then future research text. I would suggest reviewing both these sections and re-writing to be sure the conclusions succinctly summarises the key points of the review for the reader. There is also text on future research in the Discussion section which should be moved to the ‘Future Directions’ section.

Author Response

Dear Editor and Referees,

Thank you for your letter Ref. Manuscript ID: IJERPH -1832629-R1, entitled “Digital help for substance users (SU): A systematic review”, and for giving us the opportunity to review and resubmit the paper.

We are very grateful to your and the reviewers’ comments and suggestions; we are deeply appreciative of your careful reading.

Detailed replies to your comments are enumerated below, with the list of modifications and integrations. We hope this revised version now satisfies the requirements for publication in your journal.

Then, we submit the revised version of paper; for clarity new portions, added or modified in response to the referees’ comments, are highlighted in the manuscript; furthermore, the tracked version of the manuscript is attached.

Thank you very much

The Authors

Reviewer 3 Report

Thank you for the opportunity to review the referenced manuscript.  The authors did a good job in its development.   The manuscript is very similar to a 2017 systematic review of mHealth interventions in the prevention of alcohol and  substance use by Kazemi and colleagues.   I see their study is not cited as a reference.  The Kazemi study can be incorporated as part of the supporting literature for this manuscript. 

Introduction: Line # 47-48 regarding "evidence that returning to substance use after a period of abstinence leads to a more rapid reappearance..." This statement should be cited.  Also, I recommend expanding briefly on the "other features of the syndrome that occur in nondependent individuals".  This statement is not very clear to the reader.  

Methods:  The authors state that the search results were limited to English, Italian and Spanish (see line # 77).  Were there any studies found in Italian or Spanish languages?  if not, I recommend mentioning it.  If they were found, I recommend a discussion around those studies.  

Results:  For the characteristics of the studies, if at all possible, I recommend adding a column that summarizes the statistics for each study listed (p. values etc., where appropriate.  Additionally, what process/tool were used to detect bias in the selected studies?

Format/Spelling:  there is a minor spelling discrepancy with the word "heroin" it appears as "heroine" in some areas. 

Thank you.

Author Response

(The authors gave the same response as above.)

Round 2

Reviewer 1 Report

I have read the new version of your article.

Thank you for the corrections.

The article is clear

Minor check of the language of your article.